# Are Cross-Border Classes Feasible for Students to Collaborate in the Analysis of Energy Efficiency Strategies for Socioeconomic Development While Keeping CO₂ Concentration Controlled?

**Roberto Araya** [1,*] and **Pedro Collanqui** [2,3]

1 Centro de Investigación Avanzada en Educación, Instituto de Educación, Universidad de Chile, Santiago 8320000, Chile

2 Ministerio de Educación del Perú, Lima 15021, Peru; matematicapcd@gmail.com

3 Facultad de Educación, Universidad Nacional Mayor de San Marcos, Lima 15081, Peru

* Correspondence: roberto.araya.schulz@gmail.com

**Abstract:** Education is critical for improving energy efficiency and reducing CO₂ concentration, but collaboration between countries is also critical. It is a global problem in which we cannot isolate ourselves. Our students must learn to collaborate in seeking solutions together with others from other countries. Thus, the research question of this study is whether interactive cross-border science classes with energy experiments are feasible and can increase awareness of energy efficiency among middle school students. We designed and tested an interactive cross-border class between Chilean and Peruvian eighth-grade classes. The classes were synchronously connected and all students did experiments and answered open-ended questions on an online platform. Some of the questions were designed to check conceptual understanding whereas others asked for suggestions of how to develop their economies while keeping CO₂ air concentration at acceptable levels. In real time, the teacher reviewed the students' written answers and the concept maps that were automatically generated based on their responses. Students peer-reviewed their classmates' suggestions. This is part of an Asia-Pacific Economic Cooperation (APEC) Science Technology Engineering Mathematics (STEM) education project on energy efficiency using APEC databases. We found high levels of student engagement, where students discussed not only the cross-cutting nature of energy, but also its relation to socioeconomic development and CO₂ emissions, and the need to work together to improve energy efficiency. In conclusion, interactive cross-border science classes are a feasible educational alternative, with potential as a scalable public policy strategy for improving awareness of energy efficiency among the population.

**Keywords:** STEM education; energy efficiency; CO₂ emissions; APEC databases; cross-border classes; sustainable development

## 1. Introduction

Education is critical for improving efficiency in the use of energy resources [1–3]. However, there is a perception among the population that reducing energy consumption has very little impact [4]. Early education could therefore be one option to help overcome the barriers to energy-saving behaviors. Indeed, it is widely acknowledged as being one of the key social tipping elements for achieving rapid global decarbonization and climate change mitigation [5]. Moreover, several multinational organizations have been working on different educational interventions. For example, the APEC Energy Working Group (EWG) has been discussing energy efficiency since 1996. This group considers that enlightening younger generations on the importance of energy throughout elementary and secondary education is vital [6]. The Organisation for Economic Co-operation and Development (OECD) Learning Framework 2030 provides a vision and certain underpinning principles for the future of education. The first challenge proposed by this framework is an

environmental one: Climate change and the depletion of natural resources require urgent action and adaptation [7]. There are, however, several challenges with including energy efficiency in early education. Effective energy efficiency education requires the integration of different school subjects. However, truly integrating science, technology, engineering, and mathematics (STEM) is a significant challenge [8]. Furthermore, if we add socioeconomic development into the mix then the level of challenge is increased. Teachers are not used to such integration [9]. They typically teach isolated subject matters, perhaps because they themselves were trained compartmentally, with large walls dividing different university departments. In addition to this barrier, the goal of including real-world problems is even more challenging. Although teachers receive training from the mathematics, science, and education departments, they never receive any training from the engineering department. Therefore, their training is more heavily focused on understanding nature, and not on building solutions. A further challenge that is posed is the idea of cooperative learning and working in teams in order to build energy-efficient solutions. In most classes, students either listen to the teacher or work on their own, even when cooperative learning has been empirically proven to produce positive learning effects. "Despite the very robust evidence base of positive outcomes, co-operative learning 'remains at the edge of school policy'" [10]. Isolation is a deeply rooted classroom practice that is very difficult to change since teachers themselves work alone with almost no interaction with their peers. Cooperation is also difficult as schools are composed of completely independent units, i.e., isolated classrooms [11]. Moreover, an additional challenge is having students from different countries learn together and share their research, as well as explaining their understanding and their solutions. This is more demanding than simply having cross-border curricula [12]. It also requires the development of collaborative skills across countries. In this sense, students need to learn to work with teams from around the world; particularly when it comes to decarbonization strategies, where regions and countries cannot work in isolation [6].

In this article, we propose and test the feasibility of the cross-border approach for dealing with these challenges. The research question of this study is whether interactive cross-border science classes with energy experiments are feasible and can increase awareness of energy efficiency among middle school students.

## 2. Theoretical Framework

The proposed approach is based on previous experiences with cross-border public classes [13,14]. After receiving extensive feedback from the APEC Tsukuba community [15], an improved version was implemented. This version included content more geared towards the socioeconomic development of developing countries, as well as better use of technology to reach sustainable development. Given that [16–18] showed that technology can have a significant impact on learning, we include some promising online features. However, online courses can be controversial. In higher education, the completion rate for MOOCs is very low and has not improved, despite six years of investment in course development and learning research [19]. According to [20] (pp. 14,900) "online education provides unprecedented access to learning opportunities, as evidenced by its role during the 2020 coronavirus pandemic, but adequately supporting diverse students will require more than a light-touch intervention." For example, online forums can build a sense of classroom community and aid learning [21], but students do not always use these tools. On the other hand, hands-on activities for early education have proven to produce slight gains in conceptual and procedural knowledge in early energy education [1]. This is despite the fact that some misconceptions regarding energy saving and carbon-emission reductions are not easily overcome, such as foliage around the house not being simply for the sake of beautification, and grasses planted on the roof of the building not being meant only for decoration [1]. The aim of this paper is to find an adequate blend of online education with face-to-face and hands-on teaching in order to achieve good levels of student understanding. Our aim is to help them find solutions that will allow for socioeconomic development

while improving energy efficiency and controlling $CO_2$ concentrations—that is, to achieve balanced economic growth that ensures environmental protection.

The approach reported in this paper has several novel contributions. The first contribution is the introduction of an innovative new type of lesson that integrates two classes simultaneously. These are known as cross-border public classes, where two classes from different countries connect online, attend a lesson synchronously, and share their learning experience and points of view. This interconnection between classes is very important for facilitating imitation and sharing ideas, therefore helping scale the solution to national and regional levels. However, this new type of lesson requires a different didactic design and a more demanding technical infrastructure. From the designers' and teachers' point of view, the lesson has to consider that there are two classes, and that at least one of them is unfamiliar to the teacher. Furthermore, they also have to consider that the two classes do not know each other. Additionally, the didactic strategy has to help establish a fruitful communication between students from two classes that have never met each other before. These are students from different cultures who live thousands of miles apart. In this particular setting, obtaining a dialogic teaching structure, where all students participate actively, is even more challenging. The goal is not to have the students passively watch a lecturer, as is the case with most online courses [19]. Instead, the goal is to have the students receive guided inquiry instructions, where they can explore possible answers, conduct experiments, make predictions, come up with explanations, get to know their classmates from a different culture, and share their results and points of view with them, as well as peer review their classmates' solutions and explanations. One way of achieving this goal is to use ConectaIdeas, a special cloud-based platform that includes a support system for teachers [16]. This is an online platform that we have adapted for cross-border public classes. All of the students connect to the platform synchronously and answer open-ended questions on computers, tablets, or smartphones. The ConectaIdeas teacher support system helps the teacher analyze the students' responses in real time. The system also summarizes the answers and generates world clouds and concept maps in real time. These are directed graphs that link the keywords used most frequently by the students. These written responses and maps complement the verbal responses given by a small sample of students. A major challenge for the teacher is to maintain engagement in both classrooms at the same time, as well as being able to listen to the different students' reactions in order to adjust the lesson in real time.

A second contribution of this study is the work that is done to address a real-world problem using APEC databases to learn about energy consumption and energy sources, as well as their trends over the last 40 years. Furthermore, the lesson promotes analysis of the relationship between energy consumption and economic development, as well as its environmental implications. APEC and other institutions have gathered a complete set of statistics on this issue. This is a very powerful source of information, which is constantly being updated. This provides lesson designers and teachers with a unique opportunity to design lessons that are connected to real-world problems and situations that are relevant to students from different countries. In this sense, comparisons between economies are facilitated enormously. It is an invaluable source of material and a powerful mechanism for engaging students.

However, given the age and interests of middle school students, it is important to connect the issue to their everyday lives and provide concrete examples that are familiar to them [2,22]. A third contribution of this study is therefore the use of simple physical models to understand the connection between energy, socioeconomic development, and $CO_2$ concentration, as well as strategies for controlling the environmental implications of increased levels of energy consumption. $CO_2$ is a complex compound that cannot be seen directly. It is invisible; it cannot be heard nor touched. In order to be able to measure $CO_2$ concentration in the air, certain instruments are required. Although the instruments themselves are not expensive, they are not readily available in schools. However, there are some simple experiments that can help visualize $CO_2$. Nevertheless, these experiments take

up precious time and involve complex coordination if all of the students are to participate actively. This coordination is made all the more difficult when there are two or more classes that are not located in the same room. In this lesson, we use a very simple physical model. It is quick and easy and requires no materials at all. Students just have to breathe in a very small space. In this case, we suggest the students cover their mouths and noses with their hands and breathe in through the small gap between their hands. The teacher just has to make sure the hands insulate the air coming from the nose and do not leave any other gaps between their fingers. After a few seconds, students can feel an increase in the heat and humidity. They can also smell the odor of their breath and experience difficulties breathing with less $O_2$ and more $CO_2$. Furthermore, with some physical activity, such as jumping, $CO_2$ concentration increases more rapidly and is immediately detected by the students in a small, closed space. This includes more heat and humidity, higher levels of $CO_2$ and therefore increased difficulty when breathing. This difficulty to breathe creates a very direct connection to the environmental problem that must be solved urgently. Moreover, the direct experience with $CO_2$ ensures that the activity is emotionally engaging and interesting for the students. They can easily relate it to everyday experiences, such as doing physical activity in a crowded classroom or a gym full of people [23,24]. All of the students can participate and give explanations as to what is going on. The physical activity is a good model of the economic activity of a country, where more $CO_2$ emissions are produced as more energy is consumed. The lesson design also includes thinking about what happens in a bedroom, which is used as a model for the earth. In this sense, students are invited to think about a bedroom after a full night's sleep in order to model the increase of $CO_2$ concentration in the earth's air. The teacher can then dramatize the need to find a solution to the increase in $CO_2$ concentrations, highlighting the fact that, unlike the bedroom, there are no windows we can open on earth to solve the problem. By doing so, the teacher can show that another kind of solution is required.

A fourth contribution of the study is the classroom observation strategy that is adopted. In standard public classes, teachers and didactic experts observe the lesson. Then a panel analyzes and comments on the lesson. In this project, observers use SmartSpeech [25], a smartphone app that allows them to record events based on a rubric and make comments. After the class, observers upload their observations and can view them on a web-based graphical interface. Using the same app, the teacher records their speech and then uploads it to the cloud. After selecting some keywords, an algorithm counts the words and the connections between them and generates a concept map. This is a directed graph, where the main concepts are located on the nodes and where arrows connect the nodes based on the frequency with which pairs of concepts appear in the same paragraph. Additionally, the direction of the arrows reflects the order in which the concepts appeared. Efficient classroom observation is critical for measuring, sharing, and giving feedback, therefore allowing efficient pedagogical practices to be scaled [26–28].

This experience builds on previous cross-border educational experiences using Information and Communications Technology (ICT) support. Cross-border public classes and lesson study are an innovative form of teacher education and teacher professional development. For example, [29,30] illustrated a cross-border lesson study using an internet-based bulletin board system (BBS) between pairs of Japanese and Australian schools. He reported a significant cultural impact on students, awakening their cultural interest in the content and developing their hermeneutic or subjective attitude towards collaboration. There are several new ICT-based features used in the APEC project reported in this paper. The lesson features a live video transmission and augmented synchronicity, with the students jumping and answering questions live and online. Cross-border public classes and lesson study are also an additional development of massive team games across classrooms [31–34]. In these activities, dozens of schools play synchronized games with one or two teams per classroom, with each team comprising approximately 10 students. Another new addition with the APEC project is the cross-border design and the lesson study cycle that is collectively developed over 20 months by a multinational team of researchers and teachers.

The cross-border lesson reported here is a further development of a previous cross-border lesson on energy efficiency [13,14]. Having received feedback from several teachers and researchers, this latest version of the lesson now includes different activities for teaching socioeconomic development and its environmental implications.

## 3. Materials and Methods

The lesson was designed with several educational goals on energy efficiency and sustainability in mind. It was also designed to establish connections with the goals of the math and science curricula. The lesson plan prescribed a duration of one hour as shown in Table A1.

Firstly, students should realize that there is a need to compare countries using per capita estimates and not only total amounts. This means that students should realize that the size of the population for each country should be considered when looking at their $CO_2$ emissions. A larger population means more emissions. Therefore, the size of the population is critical. During the lesson, the teacher should help students discover this critical factor. During the implementation of this lesson, the students managed to detect this variable. For example, they suggested agreeing on a sample size and then taking a sample of the population of that fixed size in each country. The $CO_2$ emissions for those samples could then be calculated and compared.

Secondly, students should also consider population growth. This is also a critical factor when studying trends in $CO_2$ emissions over the last 50 years. The teacher should conduct the lesson so that students discover that a proportion of the growth in $CO_2$ emissions may be due to population growth. They should therefore look for ways to discount this.

Thirdly, students should sense $CO_2$ concentration by themselves and connect it to their everyday lives. The teacher should help them take very basic measurements by having them breathe in a very small, closed space. After a while, students should sense a change in the temperature and humidity, as well as experience greater difficulties when breathing.

A fourth goal was to learn to use modeling [8,35]. This is a critical skill. It is one of the main goals mentioned in the US Next Generation Science Standards, as well as in the Common Core Standards for Mathematics. It is a component of core science and engineering practices. In this lesson, the teacher helps the students model the level of economic activity of a country using the rhythm of their jumping, while also modeling the effects of socioeconomic development on $CO_2$ concentration by having them consider the quality of the air in their hands while jumping. The students should realize that jumping increases the temperature, humidity, and $CO_2$ concentration. They should notice the rise in $CO_2$ concentration based on the increased difficulty of breathing. Another modeling activity was to consider their bedroom as a model of the earth. They have to imagine the air quality and $CO_2$ concentration in their bedroom after a full night's sleep with closed doors and windows. The teacher then asks for possible strategies for cleaning the air, before asking whether these strategies could also be applied for cleaning the air of the whole earth.

A fifth goal was to have students think about the environmental consequences of developing countries reaching a quality of life similar to that of a developed country. Students should give a rough estimate of energy consumption per capita, total energy, and total $CO_2$ emissions if all countries were to reach the levels of socioeconomic development of a developed economy. Students should then realize the need to produce and consume energy more efficiently. They have to realize that the entire world wants to live like they do in developed countries. An increase in efficiency is therefore a must in order to reach sustainable development.

A sixth goal was that students learn to develop critical thinking [36] and be able to collaborate, share explanations and predictions, inquire, understand other students' points of view, take turns, summarize, write explanations, and do peer review with classmates from other countries. This is a significant challenge. According to the outstanding educational researcher L. Cuban [37], "writing about inquiry is far easier than practicing it in

classroom lessons." In a recent study, educational expert D. Willingham [38] concludes, "it is no surprise that programs in school meant to teach general critical thinking skills have had limited success."

Following traditional Japanese lesson studies [39], the cross-border team came to an agreement and produced a detailed lesson plan. The observers in both countries also received the lesson plan in order to track and record the class events. The lesson plan is shown in Table A1, which is located in the Appendix A. It has three columns: one with the time (from 0 to 60 min), another with the teacher's actions, and a third column with a prediction of the students' reactions.

Two sessions were held (Figure 1). The first session was taught in Lima, Peru, with a class from Santiago, Chile, also connected. The second session was taught in Santiago, with a class from Lima also connected. The first session was a first pilot whose objective was to test the idea of a cross-border class and to test the planning of the lesson. In the second session the students went from other schools. In this work we report the second session. A total of 33 eighth grade students participated, 20 from Chile and 13 from Peru. 21 were female and 12 were male. Median age was 14 years old.

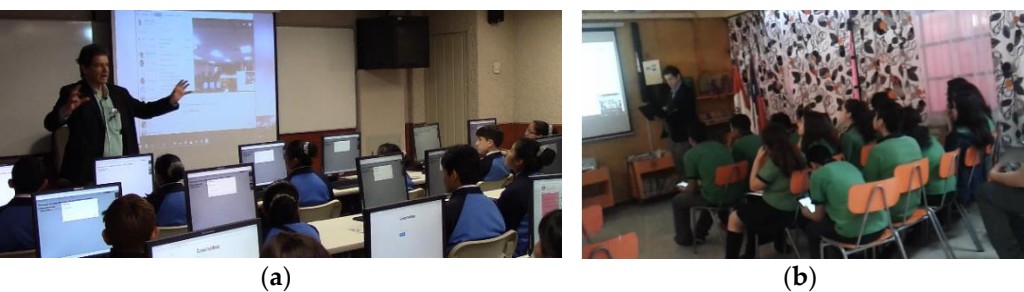

(**a**)          (**b**)

**Figure 1.** (**a**) Cross-border public class taught in Lima. In the classroom, Peruvian students are paying attention to the teacher. The Chilean class is connected by Skype and projected on the screen, along with the answers submitted on the ConectaIdeas platform by the Peruvian and Chilean students. (**b**) Cross-border public class taught in Santiago. The Chilean students answer on tablets, while the Peruvian students answer on computers (not shown).

After the teacher had the students present themselves, he then introduced the goal of learning about energy production and consumption, as well as the environmental implications. He introduced the notion of air quality and $CO_2$ concentration. At this point, the teacher introduced a first model. He made the students breathe in with their nose between their two hands, producing a small, closed volume of air surrounding the nose. The teacher asked each student to take 10 breaths and then asked what they sensed (Figure 2).

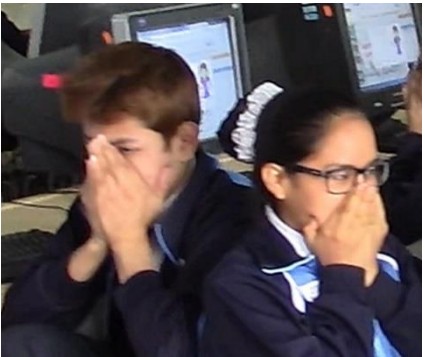

**Figure 2.** Peruvian students breathing through their nose into the small space between their hands.

The students answered that there was an increase in temperature and humidity, as well as a decrease in the air quality. The teacher then told them that this was due to an

increase in the concentration of $CO_2$ in the small volume of air surrounding their noses. He then showed Figure 3a and read question 1: *In which country have $CO_2$ emissions grown faster, and what might the cause have been? Explain your answer.* The teacher mentioned that the graphs corresponded to real data from Chile and Peru, but that he would not reveal which country was which. This is a didactic strategy to capture interest, but also a strategy to avoid comparisons of superiority between the countries. The teacher also mentioned that energy is measured in a precise and standard way, but that the graphs did not include the name of the units of energy. These students most likely did not know the exact units for measuring energy. The didactic strategy was therefore to concentrate on the socioeconomic and environmental trends rather than on concepts from physics related to energy. The teacher made sure the students understood the graph by asking for the meaning of the axes and asking for the $CO_2$ emissions for both countries at different dates. The students submitted their answers on their computers and the teacher received them in real time. The teacher then asked how the two countries should be compared. Following a discussion, the students realized that the size of the population was an important factor for explaining the amount of $CO_2$ emissions. They then suggested using equal sized samples of the population for both countries.

The teacher then showed a second slide with Figure 3b. The graph plots trends in per capita $CO_2$ emissions for both countries. On the platform, the teacher then asked question 2: *In which country did per capita $CO_2$ emissions grow faster, and what might the cause have been? Explain how you came to your conclusion.*

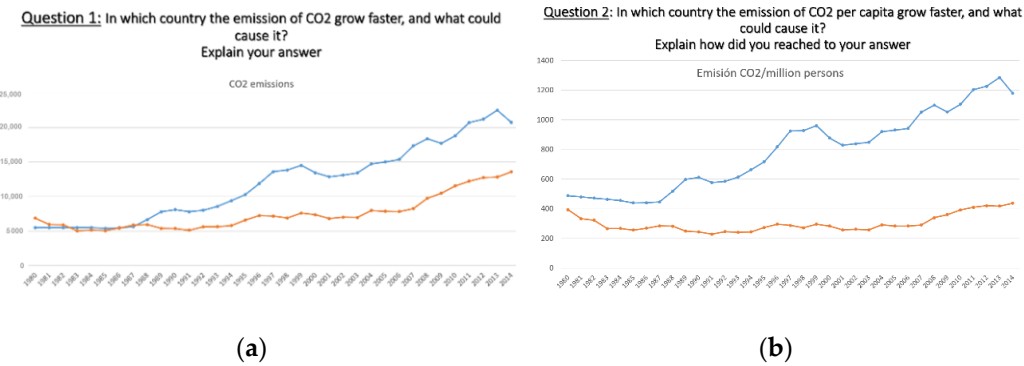

(**a**)           (**b**)

**Figure 3.** (**a**) $CO_2$ emissions and (**b**) $CO_2$ emissions per million people from 1980 to 2014 in both countries, and the corresponding question 1 and question 2 that students had to answer on their devices.

The students answered on their devices. After reviewing and commenting on the answers they received, the teacher then suggested that the two graphs could be used to calculate the population of both countries. This was a great opportunity to test the notion of division where the denominator is a rate. The teacher therefore asked question 3 on the platform: *Based on the previous graphs, in which country has the population grown faster? Explain in your own words.* The results can be easily interpreted by students since at that age they know the size of the population of their country. With this information they can work out which country is which. Some of the students realized this and mentioned it in their written responses.

The teacher then introduced the relationship between energy consumption and $CO_2$ emissions. First, he showed the graph on the left in Figure 4a and asked question 4: *In which country has energy consumption per person grown faster, and what might the cause have been? Explain your answer.* The teacher and students also commented on the trends in total $CO_2$ emissions. The teacher then suggested that the students jump 10 times while holding their hands around their noses and sensing what happened. The students realized that with more energy (jumps) the temperature, humidity, and $CO_2$ concentration increased more quickly. By doing so they could relate energy consumption and socioeconomic development with $CO_2$ concentration.

The teacher then showed the graph from Figure 4b and asked question 5: *Based on the previous graphs, compare the $CO_2$ emissions per unit of energy consumed. Explain your reasoning.* The plan was for the students to realize that it was an almost constant rate, similar for both countries, and that they have not changed much in the last 50 years. However, given time constraints this important topic was not analyzed in depth.

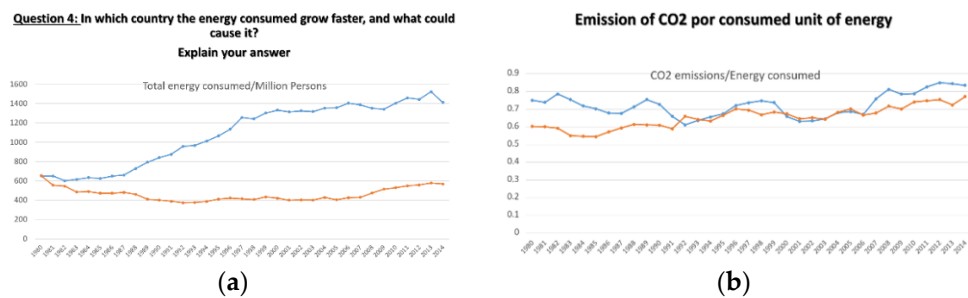

(**a**)                                                (**b**)

**Figure 4.** (**a**) Total energy consumed/million people and the corresponding question 3. (**b**) $CO_2$ emission/energy consumed and question 4 that the students had to respond to in writing on their computers or notebooks.

Next, the teacher asked the students to analyze Figure 5a [40]. This is a trend from the last 16,000 years. The students realized that there was a significant change. The teacher asked for the source of this change: What are the sources of energy and energy consumed by hunter gatherers, and also during the Roman Empire, why did it then decrease during the middle ages, and why has it skyrocketed in recent years? The students referred to electrical devices, transportation, and heating. The teacher then asked them to analyze buildings, streets, highways, and bridges, as monuments of captured energy. After that, the teacher showed Figure 5b [41]. The students realized it was another time scale, and then commented on the peak in $CO_2$ concentrations. They analyzed the source of this peak and its relation with the peak in energy consumed per capita.

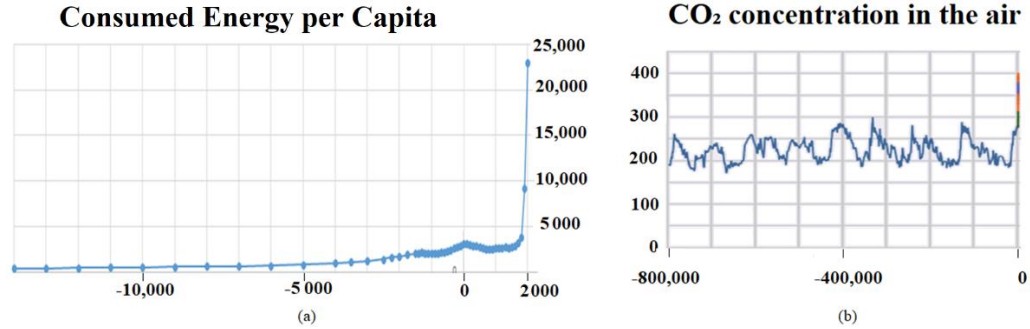

(a)                                                (b)

**Figure 5.** Graphs and the corresponding questions: (**a**) question 3, (**b**) question 4. Students had to respond in writing on their computers or notebooks. (**a**) is adapted from [39]. (**b**) is adapted from [40].

The teacher then introduced the bedroom model. He asked the students to imagine what happens to the air quality after a night of people sleeping with the door and windows closed. The students answered that there would be an increase in temperature and $CO_2$ levels. The teacher asked for the solution. The students suggested opening the windows. The teacher then asked what would happen if there were more people in the bedroom, doing physical activity like a workout in a gym. Students answered that $CO_2$ concentrations would be very high. The solution again was to open the windows. The teacher then asked if this solution could be applied to Earth. The students realized that there are no windows in this case and that another type of solution would be needed. The teacher then asked question 6 on the platform: *If every person in the world consumed as much energy as people*

*from developed countries, then how much would $CO_2$ emissions increase, what environmental implications would this have, and how do you suggest we solve it?*

## 4. Results

How do we measure the impact on 21st-century skills? Some of the data we obtained through the ConectaIdeas platform can help us measure the students' reasoning, writing, communication, and collaboration skills. For example, the platform can tell us the percentage of students who answered open-ended questions, the average length of their responses, the meaning of the their responses based on the frequency with which they mention core concepts and the connections they establish between said concepts, the percentage of students who wrote a review of another student's response, the percentage of students who received feedback, and the percentage of students who included words related to the learning goals for the lesson (i.e., normalize populations, use of models, making predictions, making suggestions, etc.).

The cloud-based platform received and summarized the written responses to the open-ended questions that were asked during the cross-border public class. For each open-ended question that was asked on the platform, the teacher received the answers from all of the students and the corresponding directed graphs with a network of the main words used by the students. A total of 29 students answered the first question, 27 students answered the second question, 28 answered the third question, 29 students answered the fourth question, and 30 students did peer review.

For the first question—*In which country have $CO_2$ emissions grown faster, and what might the cause have been?*—all of the students recognized that the blue country had grown faster. They correctly interpreted the graph. This is illustrated by the concept map in Figure 6. Some of the answers submitted by the students included:

- *The blue country grew more. This is due to the manufacturing of electronic equipment and the emission of carbon dioxide.*
- *The blue country. This is due to the fact that there are factories nearby, as well as the smoke from the cars that travel every day.*
- *$CO_2$ is growing in the blue country, this is because the population may be larger, or it may also be that it has less vegetation.*
- *In the blue country because there are probably more people in that country.*
- *The fastest growing $CO_2$ emissions are from the country represented by the blue line. This may be because it is a very industrialized country and has many factories, polluting the environment by dumping waste, etc.*
- *In Chile because there is more technology and more carbon dioxide is released into the air, because the more technology we have, the less oxygen there is.*

As can be seen from the answers above, the students gave explanations related to industrialization, transportation, vegetation, and population. These answers and explanations were in line with the predictions made in the lesson plan. These predictions can be found in the right-hand column of the lesson plan, titled "Predicted reaction of the students." The prediction was: From the graph, students will identify in which country $CO_2$ emissions have grown faster. They also proposed different causes, such as population growth and industry. However, it is interesting to note that only one student identified that the blue country was Chile.

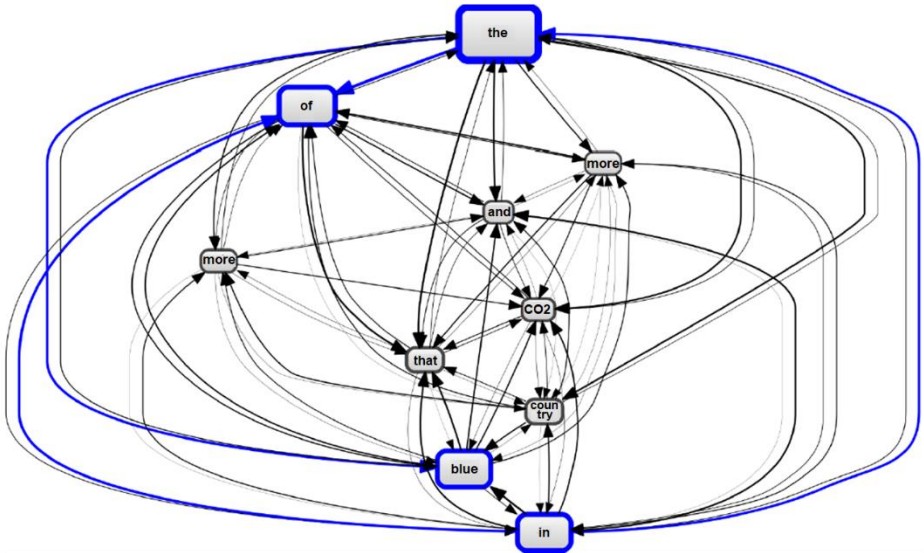

**Figure 6.** Auto-generated directed graph that summarizes students' answers to the first open-ended question. The main nodes are "blue," "country," "more," and "$CO_2$" The map basically suggests that the blue country has more $CO_2$.

However, the written explanations given by the students did not include any mention of development or economic growth. They also failed to mention the growth in transportation, industry, and population in the 34 years between 1980 and 2014. All of their explanations used only static variables. The lesson plan did not highlight this potential issue. During the session, the teacher did not pay any attention to this issue either. This important issue was only detected once the lessons had been taught, and following an in-depth review of the written responses.

The teacher asked the students how they could compare the two countries, especially given the difference in the size of their populations. After some discussion, a couple of students suggested taking the same size sample for both countries. This then led the teacher to agree on a sample of one million people. However, it was not clear whether all of the students clearly understood the meaning of this step. Following this, the teacher then asked the second question: *In which country did per capita $CO_2$ emissions grow faster, and what might the cause have been? Explain how you came to your conclusion*. Some of the written responses included:

- *In the blue country. This is because people in that country use more modern electronic devices which produce $CO_2$ and when they get rid of them, they throw them in the garbage or burn them, polluting and producing more $CO_2$, compared to the other country, which may be more rural and not use so many electronic devices . . .*
- *The blue country, maybe because there are more people.*
- *The blue one, Chile, because it has used more electricity than the other country these past few years.*

The lesson plan made two predictions regarding the students' reactions. One prediction was that students would look for ways to compare, for example, using a sample, and then understand what $CO_2$ per capita is and how it grows. The second prediction was that students would realize that in one of the two countries the per capita emission was fixed, so the total increase in $CO_2$ could be due to population growth. However, per capita $CO_2$ emissions were increasing in the other country. Most of the answers seemed to agree with these predictions. However, the second answer revealed that this student still did not understand that the information on the graph was per capita or per million people. On the other hand, it is interesting to see how one student gave a more dynamic explanation (the last of the answers shown above). This response mentions increased

electricity consumption over the "past few years." It was also this student who successfully identified the country as being Chile.

The plan was then to ask question 3 on the platform: *Based on the previous graphs, in which country has the population grown faster? Explain in your own words.* However, this question was only asked verbally. Instead, the teacher asked question 4 using the platform: *In which country has energy consumption per person grown faster, and what might the cause have been? Explain your answer.* Some of the answers included:

- *The blue country, maybe because some people use electrical appliances for a long time and therefore use a lot of electricity.*
- *The blue one, because the more energy produced per person the more $CO_2$ there is; each person generates more $CO_2$ from different activities such as driving cars, using electronic devices, etc. In addition, there are not many green spaces in that country that can purify the dirty air that comes from industrialization and pollution.*
- *The blue country, because there may be a lot more people.*
- *The blue country, because in the year 1987 the consumption of energy per person begins to increase because they start to use devices.*

All of these responses used static variables, not variations, except for the fourth one. The third answer shows that this student still did not understand that the population sizes had been normalized.

The next question from the lesson plan was only asked verbally. The next question asked on the platform was therefore, *If every person in the world consumed as much energy as people from developed countries, then how much would $CO_2$ emissions increase, what environmental implications would this have, and how do you suggest we solve it? Explain your reasoning and peer review your classmates' responses.* Some of the students' responses included:

- *In my opinion, I would say that if we were to become like the United States we would emit 100% more $CO_2$, there would be much more pollution, people would begin to disappear, so we must promote a "culture of no pollution" in each individual. At the same time, we must plant more green areas and trees so that they can purify the polluted air.*
- *It would increase the amount of pollution by three times, destroy part of the ecosystem, the ozone layer and bring respiratory diseases, so I would suggest planting more trees and making people aware of the consequences it would bring.*
- *The emissions would be three or four times the current levels, to avoid it we would have to make sure that places with lots of people are well ventilated, to try to use vehicles that produce less energy, such as bicycles, to increase the amount of green spaces, to avoid landfills and the burning of garbage.*
- *About 100 times more because the United States is a country that consumes a lot of energy and if all countries start to consume as much, it would increase drastically because there are a lot of countries in the world; all of this would have huge effects, there would be less clean air, more deaths, more diseases, hospitals full of sick people, less room per person, less nature, etc . . . We would have to have more green areas, use bicycles instead of cars, use the stairs instead of elevators, read and draw instead of watching TV or using the computer, which has actually led to many people losing the habit of reading, etc.*
- *Emission levels would be a million times higher. You can look for other ways to use energy, such as inventing a type of transport that would not emit $CO_2$, etc.*
- *The emission of $CO_2$ worldwide would be in the millions, due to the high levels of energy consumption from electrical appliances. This could be solved by taking precautions such as instead of using cars, using bicycles, forbidding factories to emit carbon dioxide, and taking care of and protecting green areas.*
- *It would explode.*

All of the students realized that $CO_2$ emissions would increase. However, they produced different estimations. Some of them just said that there would be more $CO_2$ emissions. Other students estimated an increase by a factor of three or four, but they did not give any justification for this estimation. In fact, it would appear that these estimations

were based on what the teacher said in class. He showed the left-hand graph from Figure 6 and commented that energy consumption per capita in Chile was about a third of the USA, whereas in Peru it was about a quarter. This suggests that there may have been a process of proportional reasoning. Other students suggested an increase by a factor of five, but again gave no justification. Other students proposed a factor of 100, while several others predicted that $CO_2$ emissions would be a million times higher. Some students even suggested that $CO_2$ emissions would "explode." None of the students gave any justification for their estimation. The teacher identified this diversity in the students' responses and tried to get them to think about their estimations. With more time, the students would probably have engaged in some form of proportional reasoning, explained their assumptions, and justified their estimations. This is something that should be considered for future versions of this lesson.

In addition to the ConectaIdeas platform, the SmartSpeech observation app [25] was also used by some of the teachers who observed the cross-border public class. Observers followed an observation protocol to select different options based on what they saw in the class, as well as writing their comments. Most observers selected the Classroom Observation Protocol for Undergraduate STEM (COPUS) rubric [42], proposed for STEM lessons by Physics Nobel Prize winner Carl Wieman and his team. The distribution of events observed is shown in the pie chart in Figure 7. Most of the time was spent lecturing (45%) and asking open-ended questions (43.2%). There are also observations relating to other events, such as the teacher moving among groups of students, conducting experiments, and waiting.

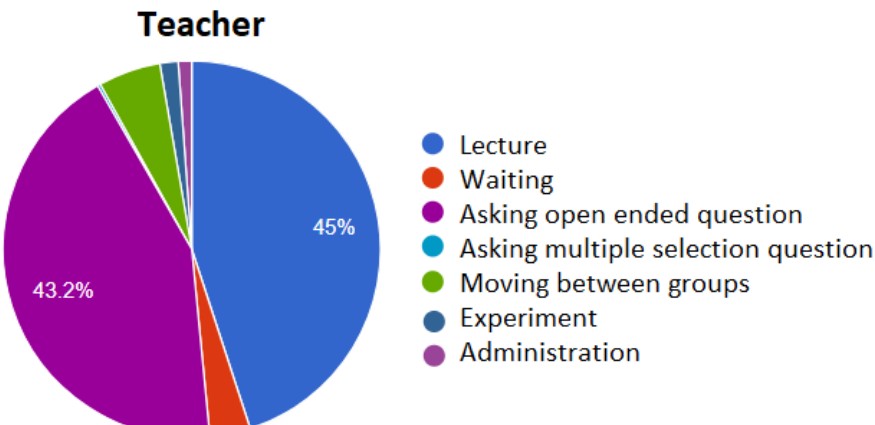

**Figure 7.** Auto-generated directed graph of the teachers' actions during the lesson. The events were marked by observers using the SmartSpeech app on their smartphone. The directed graph is a concept map summarizing the observers' selection of events according to a specific rubric, in this case the COPUS rubric.

SmartSpeech also provides a timeline of events, with the duration and occurrence of each event (Table A2).

SmartSpeech displays the time schedule graphically (Figure 8). In this case, we can see that experiments were conducted at two different points during the lesson. These events correspond to the breathing activities.

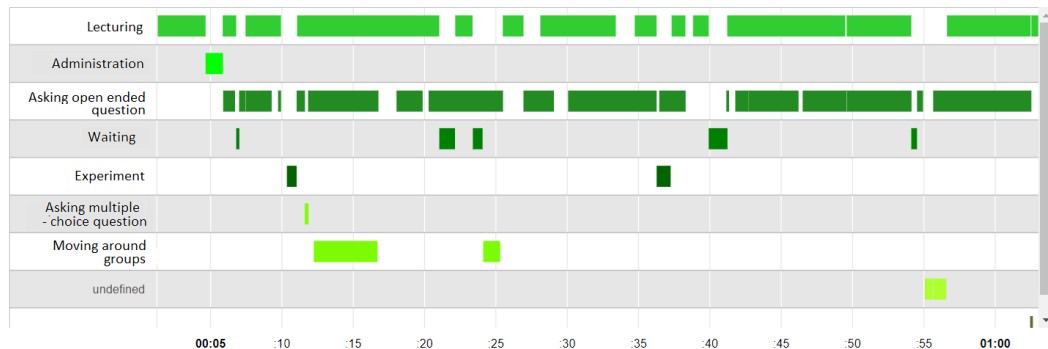

**Figure 8.** Auto-generated time graph summarizing the events recorded by the observers. On the horizontal line is the time. The graph displays the events labeled in the first column as horizontal bars according to the time of their occurrence.

During the class, the teacher recorded his speech on his smartphone using the Smart-Speech app. Once the lesson was finished, the app uploaded the recording to the cloud and used Google services to produce a transcript of the recording based on automatic speech recognition. The teacher then selected some of the main concepts that were used most frequently, with SmartSpeech generating a corresponding concept map (Figure 9). In this case, we can see how the concept of energy was related to more heat, growth, and odor. Additionally, electricity and transportation were also linked to the concept of energy. All of these issues were connected to the idea of "more energy," which is the main arrow on the map. This was the concept that summarized the whole lesson.

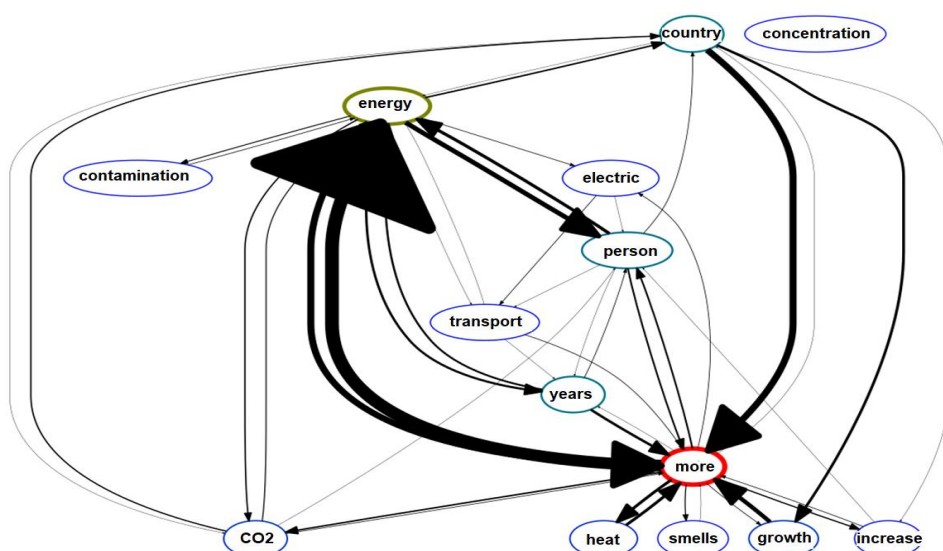

**Figure 9.** Auto-generated concept map summarizing the main concepts from the teacher's discourse and the relationships between them. It basically suggests that a country with more energy impacts contamination and $CO_2$ emissions, but also that countries and people generate more heat, odor, and growth.

The use of these electronic devices and software helped with the implementation of the cross-border public class. The ConectaIdeas platform and the SmartSpeech App were combined to create a performance support system [43] that helped the teacher deliver a cross-border public class [16]. It allowed the teacher to contact all of the students, receive their written responses to open-ended questions, and receive a concept map as a summary while they were teaching.

## 5. Discussion

Lesson study and public classes are a very powerful strategy for delivering professional development to teachers. It is a Japanese teacher education and training strategy that has been around for more than 150 years. For several years, APEC economies have been developing STEM lessons based on critical, real-world problems, such as emergency preparedness for events such as tsunamis, hurricanes, earthquakes, landslides, and forest fires [44]. In this paper we report on the design and implementation of a lesson on a new problem: energy efficiency and decarbonization. There was also the additional challenge of designing a cross-border public class in order to help students develop interpersonal skills and learn to cooperate with peers from other countries, as well as developing cross-cutting concepts related to energy, social development, and the environment. The goal was to tear down the walls that separate and isolate classrooms, and bring them closer to what is happening in the real world. Using computers and mobile devices, we can enable the shift from isolated technology labs to technology-rich classroom contexts [45].

Did this cross-border public class meet its goals? From the written responses that were submitted we obtained some evidence of student achievement. We saw how the lesson helped students understand the cross-cutting nature of energy. Several of them connected it to concepts from physics, such as heat and movement (kinetic energy). They also wrote about the transformation of energy, such as going from heat or fuel to movement. In addition to this, they also made connections with the concept of energy from biology, such as metabolism, calories, diets, and food. Finally, they also made connections to socioeconomic development, for example, comparing transportation, electricity consumption, and fuel use in the two countries, as well as looking at trends from the last 40 years. Students also related energy to $CO_2$ emissions.

Another goal was to help students think critically. According to [36], this is a central goal of science education: Students should learn to think critically about scientific data and models. In this sense, the authors claim that this ability is developed through repeated practice by making decisions based on data and receiving feedback on those decisions. In this cross-border public class, the students reviewed APEC databases on energy production and consumption, as well as $CO_2$ emissions. They had to understand the information, its graphical representations, and the different temporal trends. They had to make meaningful comparisons between countries, understand the implications and share their views. They also had to write down their observations, explanations, and suggestions, and they received feedback from their peers, some of whom were from another country. Therefore they had to learn to listen, read, and respond in writing, commenting on the suggestions and opinions of students from another country. Moreover, they had to use a couple of models to understand the experimental data in more depth and make predictions. As suggested by the Next Generation Science Standards (NGSS) and several authors [35], models provide scientists and engineers with tools for thinking. Such models can help them visualize and make sense of different phenomena and experiences, before developing possible solutions to design problems. The framework for K–12 science education [8] states that by the end of 12th grade students should be able to represent and explain phenomena using multiple types of models and move flexibly between model types depending on their purpose and use. In this cross-border public class on energy efficiency students used several models and discussed their respective strengths and weaknesses.

However, students were still missing some critical concepts. Firstly, not all students realized that there was a need to normalize the data. Even when the teacher displayed information on emissions and electricity consumption per million people, some students did not appreciate its significance and importance for cross-country comparisons, as well as for understanding causal mechanisms that could explain temporal trends. Secondly, most of the students only gave argumentations using static variables in response to the temporal patterns they identified. For the increase in emissions or electricity consumption over the 35-year period under study they only mentioned the current state of important factors in the respective countries, but not the changes in those factors over the 35 years.

More specifically, they did not explicitly consider the impact of socioeconomic growth. Moreover, given that the global emission intensity (fossil fuel $CO_2$ emissions per GDP) rose in the first part of the 21st century, socioeconomic growth plays a more significant role than could be predicted by just proportional reasoning. Previous emission intensity projections failed since they did not consider unanticipated GDP growth in Asia and Eastern Europe [46]. Thirdly, students did not provide any justification for their predictions when asked how much $CO_2$ emissions would increase if all countries lived like developed economies. Some students gave reasonable estimates, probably based on proportional reasoning. However, they did not justify their arguments. They did not explain their assumptions or the proportional reasoning underpinning their estimations. Finally, the majority of students did not consider or explicitly argue for the need to improve energy efficiency. These are issues that should be considered for future versions of the lesson.

There is also another important topic that teachers may wish to include in future versions of the lesson: the Jevons paradox [47]. This paradox is critical for understanding the environmental impact of technological transformations that generate more efficiency. According to this paradox, an increase in efficiency leads to an increase in consumption, not the other way around. This is completely counterintuitive. In 1865, William Jevons observed that a more efficient use of coal led to an increase in its consumption. Watt's innovations with steam engines made coal a more cost-effective power source and therefore it started to be used in more industries. This is perhaps the most widely known paradox in environmental economics. It would certainly engage students in a deep discussion that should be highly relevant to citizens of developing countries. Doing so may provide them with the necessary awareness and knowledge they need to face the challenges of energy consumption and the environment in the 21st century.

## 6. Conclusions

In summary, we can conclude that the main objective of the lesson was met. We found that cross-border classes are a viable option. We also found that even though the two classes were far apart from one another, all of the students in both classes participated actively. Students were able to collaborate in seeking solutions together with others from the other country. In addition, we found that it is possible to run an online cross-border class that is not merely expository, but instead active and focused on inquiry and experimentation. We also found that it is possible to develop verbal and written argumentation in all students. Furthermore, we also discovered that synchronous peer review is entirely possible, with students reviewing and commenting on each other's written responses in real time. Additionally, it turned out to be a very attractive class for the students, with students from two countries working together to explore solutions to the issue of promoting socioeconomic development while also keeping $CO_2$ emissions in check. That is, students from two countries worked to imagine and discuss strategies that achieve economic growth that is compatible with sustainable development. These preliminary findings are encouraging signs that with middle school education it is possible to change the current perception among the population that reducing energy consumption has very little impact.

**Author Contributions:** Conceptualization, P.C. and R.A.; data curation, R.A.; formal analysis, R.A.; funding acquisition, R.A.; investigation, P.C. and R.A.; methodology, R.A.; project administration, R.A.; software, R.A.; supervision, P.C. and R.A.; validation, P.C.; visualization, P.C. and R.A.; writing—original draft, R.A.; writing—review and editing, R.A. All authors have read and agreed to the published version of the manuscript.

**Funding:** This work was supported by the Chilean National Agency for Research and Development (ANID), grant number ANID/ PIA/ Basal Funds for Centers of Excellence FB0003.

**Institutional Review Board Statement:** Ethical review and approval were waived for this study, due to it being a class session during school time. The activity was revised and authorized by the respective principals.

**Informed Consent Statement:** Student consent was waived due to authorization from principals. Given that there are no patients but only students in a normal session in their schools, within school hours, and using a platform that records their responses anonymously, the principals authorized the use of anonymized information and the pictures for publication.

**Data Availability Statement:** The minimum data set that supports findings is shared in Appendix A, figures, and anonymized student responses.

**Acknowledgments:** Support from ANID/PIA/Basal Funds for Centers of Excellence FB0003 is gratefully acknowledged.

**Conflicts of Interest:** The authors declare no conflict of interest. The funders had no role in the design of the study; in the collection, analyses, or interpretation of data; in the writing of the manuscript, or in the decision to publish the results.

**Appendix A**

**Teacher Goals:** Give students the opportunity to

- Reflect on the cross-cutting concept of energy and $CO_2$ emissions and its connections with the need for improved energy efficiency; and
- Connect with students from different schools and countries to reflect together on the need to improve energy efficiency.

**Goals for students:** to recognize the cross-cutting concept of energy, its relationship to $CO_2$ emissions and socioeconomic development, and recognize the need to work together to improve energy efficiency to reach sustainable development.

**Mission:** to help achieve social development across the world with greater energy efficiency.

**Materials:** graphics, ConectaIdeas platform.

**Table A1.** Lesson plan for the cross-border public class on energy efficiency.

| | Teacher Activities | Predicted Reaction of Students |
|---|---|---|
| 0–15 min | **Stage 1: Getting to know each other and answering the first question on the platform**<br>• The teacher asks the students to breathe in their hands and asks about air quality.<br>• The teacher shows a graph with the $CO_2$ emissions from two countries and asks in which country the $CO_2$ emissions are growing faster and why.<br>• The teacher chooses some of the answers from the platform and comments on them. | • Students describe the temperature changes in their hands and the air quality, but they do not necessarily relate this to $CO_2$.<br>• From the graph, students detect in which country $CO_2$ emissions have grown faster. They propose different causes such as population growth and industry. |
| 15–30 min | **Stage 2: $CO_2$ per capita**<br>• The teacher asks how to compare the countries when they have different populations.<br>• The teacher shows a second graph showing the per capita $CO_2$ emissions for the two countries and asks which country's per capita $CO_2$ emissions are growing faster and what the reason might be.<br>• The teacher asks about population growth in both countries. | • Students look for ways to make comparisons, for example using a sample, and then understand what $CO_2$ per capita is and how it increases.<br>• They realize that in one of the two countries the per capita emissions are fixed, so the total increase in $CO_2$ could be due to population growth. However, per capita $CO_2$ emissions are increasing in the other country.<br>• Students (with some help) calculate the population of both countries, both at the beginning and at the end of the period, and realize that the country with the lowest $CO_2$ emissions is growing the most in terms of its population. |

**Table A1.** *Cont.*

|  | Teacher Activities | Predicted Reaction of Students |
|---|---|---|
| 30–45 min | **Stage 3: Energy consumption:**<br>● The teacher asks the students to calculate the population in both countries and compare the growth of total emissions with population growth.<br>● The teacher has the students jump and asks them to breathe in their hands and then asks them about the air quality.<br>● The teacher suggests comparing emissions graphs with economic activity (energy) graphs, and asks the students which are the main industries in each country, as well as their energy sources (from fire, to coal, to gas, and now to solar energy).<br>● The teacher shows the students the graph for total energy consumption and total energy per capita and asks about the rate of increase and whether there are similar trends on the $CO_2$ emission graphs. | ● The students realize that more movement leads to a decrease in the air quality and that this is analogous to greater economic activity.<br>● The students comment on their country's main industries.<br>● They are aware of similar trends in energy (total and per capita) and in $CO_2$ emissions. |
| 45–60 min | **Stage 4: Energy efficiency**<br>● The teacher asks the students how to calculate the efficiency of the $CO_2$ emissions.<br>● The teacher shows the students the graphs for $CO_2$ emissions per unit of energy consumed and asks if this is similar in both countries.<br>● The teacher shows the students a graph for the energy captured per capita per day since 15,000 B.C. and shows where the two countries are on the graph, as well as other, more developed countries.<br>● The teacher shows the students a graph of $CO_2$ concentration in the air over the last 800,000 years and asks for their interpretation, as well as to predict how much $CO_2$ emissions would increase if everyone lived like people in developed countries. The teacher also invites the students to suggest possible solutions.<br>● The teacher makes sure that students peer-review their classmates' responses. | ● Students describe efficiency as cleaner air but without reducing energy.<br>● Students realize that the $CO_2$ emissions from energy consumed are similar in both countries and have remained similar, and surprise that it is a similar proportion in all countries.<br>● Students predict much higher $CO_2$ emissions in the United States.<br>● Students predict an explosion of $CO_2$ emissions if all countries live like they do in the United States. They propose cleaner energy, such as wind and solar. |

**Table A2.** Time schedule of events using the COPUS rubric.

| Start | End | Duration | Teacher Action |
| --- | --- | --- | --- |
| 0:01:14 | 0:04:39 | 0:03:25 | Lecture |
| 0:04:39 | 0:05:52 | 0:01:13 | Administration |
| 0:05:51 | 0:06:48 | 0:00:57 | Lecture |
| 0:05:53 | 0:06:43 | 0:00:49 | Ask questions |
| 0:06:48 | 0:07:01 | 0:00:12 | Waiting |
| 0:07:00 | 0:07:27 | 0:00:27 | Ask questions |
| 0:07:27 | 0:10:03 | 0:02:35 | Lecture |
| 0:07:28 | 0:09:17 | 0:01:49 | Ask questions |
| 0:09:44 | 0:10:02 | 0:00:18 | Ask questions |
| 0:10:21 | 0:11:02 | 0:00:41 | Experiment |
| 0:11:02 | 0:11:37 | 0:00:34 | Ask questions |
| 0:11:04 | 0:21:02 | 0:09:57 | Lecture |
| 0:11:36 | 0:11:52 | 0:00:16 | Ask multiple-choice question |
| 0:11:50 | 0:16:46 | 0:04:56 | Ask questions |
| 0:12:14 | 0:16:42 | 0:04:28 | Move around student groups |
| 0:18:02 | 0:19:52 | 0:01:50 | Ask questions |
| 0:20:17 | 0:25:30 | 0:05:13 | Ask questions |
| 0:21:02 | 0:22:08 | 0:01:06 | Waiting |
| 0:22:09 | 0:23:22 | 0:01:13 | Lecture |
| 0:23:23 | 0:24:04 | 0:00:41 | Waiting |
| 0:24:07 | 0:25:18 | 0:01:10 | Move around student groups |
| 0:25:29 | 0:26:56 | 0:01:26 | Lecture |
| 0:26:56 | 0:29:04 | 0:02:08 | Ask questions |
| 0:28:07 | 0:33:17 | 0:05:09 | Lecture |
| 0:29:59 | 0:36:16 | 0:06:17 | Ask questions |
| 0:33:18 | 0:33:19 | 0:00:01 | Lecture |
| 0:34:44 | 0:36:16 | 0:01:31 | Lecture |
| 0:36:16 | 0:37:15 | 0:00:58 | Experiment |
| 0:36:27 | 0:38:18 | 0:01:50 | Ask questions |
| 0:37:19 | 0:38:17 | 0:00:58 | Lecture |
| 0:38:49 | 0:39:55 | 0:01:06 | Lecture |
| 0:39:55 | 0:41:14 | 0:01:18 | Waiting |
| 0:41:13 | 0:49:29 | 0:08:15 | Lecture |
| 0:41:14 | 0:41:16 | 0:00:02 | Ask questions |
| 0:41:47 | 0:42:43 | 0:00:55 | Ask questions |
| 0:42:43 | 0:46:13 | 0:03:29 | Ask questions |
| 0:46:30 | 0:49:35 | 0:03:04 | Ask questions |
| 0:49:35 | 0:54:07 | 0:04:31 | Lecture |
| 0:49:36 | 0:54:08 | 0:04:31 | Ask questions |
| 0:54:07 | 0:54:31 | 0:00:23 | Waiting |
| 0:54:31 | 0:54:59 | 0:00:27 | Ask questions |
| 0:54:57 | 0:55:39 | 0:00:42 | |
| 0:55:39 | 1:02:32 | 0:06:53 | Ask questions |
| 0:55:41 | 0:56:34 | 0:00:52 | |
| 0:56:37 | 1:02:29 | 0:05:52 | Lecture |
| 1:02:32 | 1:03:01 | 0:00:28 | Lecture |
| 1:02:32 | 1:02:33 | 0:00:00 | Answer questions |

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
