# Peer review of "Are Cross-Border Classes Feasible for Students to Collaborate in the Analysis of Energy Efficiency Strategies for Socioeconomic Development While Keeping CO2 Concentration Controlled?"

_sustainability, doi:10.3390/su13031584_

Round 1
Reviewer 1 Report
Dear Authors:
I think the manuscript is interesting and withing the scope of this special issue. However, I have some concerns that I do think could make the manuscript more appealing to readers:
1) I recommend improving abstract section, for instance to make it clear the research objectives and a highlighting briefly the main results obtained.
2) Introduction section: Could authors add some references about the statement “On the other hand, hands-on activities for early education have proven to produce slight gains in conceptual and procedural knowledge in early energy education. This is despite the fact that some misconceptions regarding energy saving and carbon-emission reductions are not easily overcome”. Also, I would appreciate some examples about these misconceptions author are indicating.
3) Introduction section: From line 104 at until the end of this section is a bit confusing. As a reader I expected to know the research goals at the end of this section, not in the middle. I would recommend rewriting and consider writing two separate parts (introduction and theoretical framework).
4) Material and methods: Basic information about the lesson is missing, for instance, duration, number of sessions, contents, … In general, this section is very confusing. Authors have included a table (Table A1), please add this table at the beginning of the lesson description in order to guide the reader. In addition, what table is named as “A1”. In addition, in material and methods, authors indicate “following a Japanese Lesson Studies” but no information about this methodology is provided in introduction section.
5) Sample information is missing.
6) Instrument information is missing. What information was getting from students? How many students have provided answers? Authors should add more information in order to understand the implication of their research.
7) Data analysis: Authors did a descriptive analysis? A qualitative analysis? Please add more information about how the study was conducted.
8) Figure 6 and 9 are in Spanish, please change to English.
9) In results section, authors indicated how software provide the results, but I do need to know more precise information (demographic information, number of students participating…). In general authors said “the students gave explanations…” but does it mean that all the students?
10) It would be desirable to split discussion in two sections: discussion and conclusion.
Author Response
We appreciate your time and dedication to review our paper. We agree with many of your comments. Your suggestions have allowed us to clarify the objective, improve the title and abstract, and clarify various points of the paper.
Attached is a response point-by-point.

Reviewer 2 Report
This manuscript deals with different interesting and important issues. First transversal topics education. Second, energy use and socioeconomic development. Third, methods or learning tools to approach those previous two. From the journal point of view this is a hot topic that fits the aims.
However, there are some aspects that need to be improved before a final acceptance.
- In your title you say “Early education”. What do you mean with this? If you talk within the text on different K-levels, at the very end you didn’t mention the level or age of students. Generally Early education refer to those ages till the 9 (first degree level). From the pictures they can be 12-15, but I’m not sure. SO, these are two weak aspects: the title that induces in error; and the missing information about the age or the level of classes.
- In line 67 you start with the objectives of this study: to test the feasibility of one approach. And at the end of the paragraph you refine the objective aiming find an adequate blend online education with face-to-face and hands-on teaching to achieve good levels of student understanding. To test that you always need a control and you had none. How then can you take out conclusions ?
These in fact are the major problems. The innovation of methodologies, the different software you tested, the apps, the hands-on approaches, are really amazing but, what were your conclusions between the two-classes face-to-face? There were differences ? Did you catch them? Did they have the same behaviour? What was the advantage to make it face-to-face ? How much did they interact in terms of grouping? And in what aspects of approaching the study problem? Did you compare with normal classroom lecturing using the same hands-on and modelling schemes? How many time did you repeat this ? was it launch once? How many weeks does this study take? The final appendices table is not clear enough. There was no home work nor even student stimulus to understand better some scientific contents?
As a minor aspects, I do think you should avoid describe the innovations in introduction section. With that description we loose your major focus of the study. In my opinion those description should be “touched” in introduction and described in material and methods.
In M&M you should define precisely what students you have, age, level of knowledge, gender ratio, as well as teachers. What were their background?
I see no need on Figure 1. and 2, although the 2 is much more informative than 1.
Author Response
Dear reviewer
Thank you very much for your time and dedication to make a detailed review of our paper. Your suggestions have allowed us to improve the paper. We have changed the title and abstract and clarified several points of the paper.
Attached is the response point-by-point.

Round 2
Reviewer 1 Report
Dear Authors: I do think the revised version is ready for its publication. I just recommend to revise English once again. Regards
Author Response
Dear Reviewer
Thank you very much.
Attached is our response.
Best regards

Reviewer 2 Report
Dear Authors
The manuscript is now much improved. The only thing I see is that your conclusion section is a too large section, though I understand your aspect. But I would advise to enlarge your discussion and then reframe your conclusions
Author Response
Dear Reviewer
Thank you very much for your time and dedication to help us improve the manuscript.
Attached is our response.
Best regards
